# A Long Polymorphic GT Microsatellite within a Gene Promoter Mediates Non-Imprinted Allele-Specific DNA Methylation of a CpG Island in a Goldfish Inter-Strain Hybrid

**DOI:** 10.3390/ijms20163923

**Published:** 2019-08-12

**Authors:** Jianbo Zheng, Haomang Xu, Huiwen Cao

**Affiliations:** 1Zhejiang Institute of Freshwater Fisheries, Huzhou 313001, China; 2College of Life Sciences, Zhejiang University, Hangzhou 310058, China

**Keywords:** allele-specific DNA methylation, CpG island, non-imprinted gene, cis-acting regulatory polymorphisms, GT microsatellites

## Abstract

It is now widely accepted that allele-specific DNA methylation (ASM) commonly occurs at non-imprinted loci. Most of the non-imprinted ASM regions observed both within and outside of the CpG island show a strong correlation with DNA polymorphisms. However, what polymorphic *cis*-acting elements mediate non-imprinted ASM of the CpG island remains unclear. In this study, we investigated the impact of polymorphic GT microsatellites within the gene promoter on non-imprinted ASM of the local CpG island in goldfish. We generated various goldfish heterozygotes, in which the length of GT microsatellites or some non-repetitive sequences in the promoter of *no tail* alleles was different. By examining the methylation status of the downstream CpG island in these heterozygotes, we found that polymorphisms of a long GT microsatellite can lead to the ASM of the downstream CpG island during oogenesis and embryogenesis, polymorphisms of short GT microsatellites and non-repetitive sequences in the promoter exhibited no significant effect on the methylation of the CpG island. We also observed that the ASM of the CpG island was associated with allele-specific expression in heterozygous embryos. These results suggest that a long polymorphic GT microsatellite within a gene promoter mediates non-imprinted ASM of the local CpG island in a goldfish inter-strain hybrid.

## 1. Introduction

It is well known that allele-specific DNA methylation (ASM) at imprinting loci induces allelic-specific expression (ASE) during the development of mammals [1,2]. There is now a lot of evidence that ASM also commonly occurs at non-imprinted loci and can result in allele-specific expression in some cases [3,4,5,6,7]. Most of the non-imprinted ASM regions observed both within and outside of CpG islands show a strong correlation with nearby DNA polymorphisms [3,4,6,7,8,9,10,11]. A significant fraction of non-imprinted ASM regions observed outside of CpG islands in human genome is dependent on the presence of heterozygous single-nucleotide polymorphisms (SNPs) at CpG dinucleotides in local regions [6,7,10]. However, it is not yet clear what polymorphic *cis*-acting regulatory elements affect ASM of CpG islands at non-imprinted loci. 

Simple tandem repeats with repeat units of 1–6 nucleotides, often referred to as microsatellites, are rich in the genomes of both eukaryotic and prokaryotic species [12,13,14,15]. Microsatellites are highly unstable and exhibit length polymorphisms with multiple alleles at both individual and population levels [16,17,18,19,20,21]. Growing evidences suggest that the length polymorphisms of microsatellites have functional importance. Length variations of several microsatellites in human genome can alter disease susceptibility [22,23]. A recent study has found that polymorphic microsatellites within gene promoters act as modifiers of local DNA methylation in humans [24]. These findings stimulated us to investigate whether the length polymorphisms of microsatellites within gene promoters have impact on ASM of the local CpG island at non-imprinting loci in vertebrates. 

Repeat dinucleotides (GT)_n_ is one of the most common microsatellites present in all eucaryotes [25,26,27]. In the promoter of *no tail* (*ntl*), there is a CpG island close to the transcription start site and a conserved GT microsatellite at the upstream region in both zebrafish and goldfish a key developmental regulatory gene. Previous study has observed that this CpG island is highly methylated in the eggs of zebrafish and a bisexual diploid goldfish strain, *C. auratus auratus*, but unmethylated in the eggs of a unisexual polyploid goldfish strain, *C. auratus pengze* [28]. During embryogenesis, goldfish and zebrafish maternally methylated *ntl* CpG islands undergo demethylation first, followed by de novo methylation of both maternal and paternal alleles [28,29], indicating that *ntl* is a non-imprinted locus. Moreover, the methylation of the *ntl* CpG island is involved in the *ntl* repression in both goldfish and zebrafish [28,29]. Interestingly, the GT microsatellite in the *ntl* promoters of both the bisexual diploid goldfish and zebrafish is much longer than that in the *ntl* promoter of unisexual polyploid goldfish [30]. Therefore, the GT microsatellites and the linked downstream CpG island in the *ntl* promoter is a suitable experimental system for our investigation. 

In this study, we screened two bisexual goldfish strains with different genetic backgrounds and generated various goldfish heterozygotes, in which the length of GT microsatellites or some non-repetitive sequences in the promoter of *ntl* alleles is different. By examining the methylation status of the downstream CpG island in these heterozygotes, we found that the length polymorphisms of this GT microsatellite can mediate the ASM of the downstream CpG island during oogenesis and embryogenesis. 

## 2. Results

### 2.1. Sequence Divergence within Upstream Region of the ntl Promoter Accompanying by Methylation Variation of Downstream CpG Island 

By sequencing and comparing the *ntl* promoter in many goldfish strains with different genetic backgrounds, we observed that two bisexual goldfish strains, a Chinese goldfish (Chinese GF) strain, *Carassius auratus auratus,* and a Japanese goldfish (Japanse GF) strain, *Carassius auratus cuvieri*, contain the desired sequence divergence in their *ntl* promoter. There are three GT microsatellite loci, named as GTM 1, 2 and 3 according to the location order from up to downstream, in their upstream region of the *ntl* promoter (Figure 1a). GTM 1 is longer than GTM 2 and 3 in each of the two strains. Both GTM 1 and 3 are longer in the Chinese GF strain than in the Japanese GF strain. The length of GTM 2 is the same in the two strains. In the flanking regions of the GTMs, there are multiple diversified non-repetitive sequences including insertion/deletions and single-nucleotide polymorphisms (SNPs) sites, which can be used as Chinese GF and Japanese GF genetic makers. Two diversified *ntl* CpG island with an 11 bp long insertion/deletion variation (including two CpG sites) were detected in the Japanese GF strain but no diversified *ntl* CpG island was detected in the Chinese GF strain. The sequence and length of the *ntl* CpG island in the Chinese GF strain are exactly the same as that of the long *ntl* CpG island in the Japanese GF strain (Appendix A). 

The previous study has observed that the *ntl* CpG island is highly methylated in the eggs but unmethylated in the sperms in the Chinese GF strain [28]. We then examined the methylation status of the CpG island in gametes of Japanese GF strains by BS and compared with that in the gamete of the Chinese GF strain. Unlike in the Chinese GF strain, no methylated clone of the *ntl* CpG island was detected in the eggs and the sperms of Japanese GF homozygote (Figure 1b and Appendix A). In Japanese GF heterozygote, only a few slightly methylated clones of the long *ntl* CpG island were detected in the eggs (Appendix A). These results suggest that the sequence divergence at the upstream region of the *ntl* promoter in the Chinese GF and Japanese GF strains is accompanied by methylation variation of downstream CpG island in the eggs. However, the insertion or deletion of the 11 bp long fragment in the CpG island has no significant impact on the methylation of the CpG island.

### 2.2. Impact of Diversified Promoter Sequences on the Methylation of Downstream CpG Island in Hybrid 

To prove that the diversified sequences in the upstream regions of the Chinese GF and Japanese GF *ntl* promoter play a role in regulating the methylation of the CpG island, we generated an intraspecific hybrid by mating a homologous Chinese GF male individual to a homozygous Japanese GF female individual. In this hybrid, the lengths of GTM 1 and 3 is 168 bp and 56 bp long, respectively, in the Chinese GF *ntl* allele, 72 bp and 36 bp long, respectively, in the Japanese GF *ntl* allele, the Chinese GF and Japanese GF *ntl* CpG island can be distinguished by the 11 bp long indel (Figure 1a and Appendix A). The hybrid developed smoothly and the mating of mature heterozygous female and male reproduced normal offspring. 

After sexual maturity achieved, we examined the methylation status of the *ntl* CpG island in the matured eggs of the hybrid by MSP and BS. The MSP experiment detected two different methylated fragments of the CpG island in the eggs of the hybrid (Appendix A). The length of the long and short MSP fragment was equal to that of the Chinese GF and Japanese GF *ntl* CpG island, respectively. In the controls, only the long MSP fragment was detected in the Chinese GF eggs. The BS analysis showed that the ratio of the methylated and unmethylated clones of the *ntl* CpG island was about one to one as a whole (Figure 1b), correlating well with mono-allele methylation in heterozygote. By distinguishing the Chinese GF and Japanese GF genetic markers in the *ntl* CpG island of sequenced BS clones, we observed that most of the sequenced Chinese GF-derived *ntl* CpG island clones were methylated clones, and a few were unmethylated clones. In contrast, most of the sequenced Japanese GF-derived *ntl* CpG island clones were unmethylated clones, and a few were methylated clones. 

Based on these observations, we have reason to speculate about the following possibilities. (1) Sequence diversification in the upstream region of the Chinese GF *ntl* promoter has generated *cis*-acting elements that can mediate the methylation of downstream CpG island in the eggs of hybrid. (2) Reciprocal DNA fragments exchange occurred between the Chinese GF and Japanese GF *ntl* alleles at the upstream region of the promoter due to meiotic homologous recombination (HR) and resulted in methylation status conversion between the Chinese GF and Japanese GF CpG island in a small number (5/20) of the eggs of the hybrid. 

### 2.3. Homologous Recombination at ntl GT Microsatellite in the Eggs of the Hybrid 

It has been observed that long GT microsatellites are HR hot spots in the human genome [31]. An insertion of (GT)_n_ repeat in a yeast chromosome can promote reciprocal DNA fragment exchange between homologous chromosomes during meiosis [32,33]. To demonstrate that reciprocal DNA fragment exchange occurred between *ntl* alleles at the upstream region of the promoter in the hybrid, we analyzed the linkage patterns of the Chinese GF and Japanese GF genetic makers flanking GTM 1 in the egg genome of the hybrid. In the PCR products amplified from the eggs of the hybrid, most of the sequenced *ntl* promoter clones were the Chinese GF and Japanese GF wild type clones (Figure 2a) but a small portion of them was HR clones (Figure 2b). No HR clones was detected in the PCR products amplified from the equivalent mixture of homozygous paternal and maternal genomic DNA (Figure 2c), indicating that the HR clones detected in the eggs of the hybrid were unlikely to be the products generated by template-switching during the PCR reaction. This revealed that frequent HR indeed occurred at the GTM 1 between the Chinese GF and Japanese GF *ntl* alleles. 

The reliability of HR between the Chinese GF and Japanese GF *ntl* alleles at GTM 1 in the hybrid was confirmed by analyzing the genotype in the meio-gynogenetic diploid progeny of the hybrid. The meio-gynogenetic diploid progeny of the hybrid was generated by inducing the development of the matured eggs of a heterozygous individual with a genetic inactivated common carp sperm and then inhibiting the second polar body release of the activated eggs. In this artificial meio-gynogenetic diploid progeny, both copies of a given chromosome are the duplicates in an individual, thereby the percentage of gynogenetic heterozygous diploid (GHD) individuals in the gynogenetic progeny group directly reflects the frequency of meiotic HR during oogenesis. By analyzing the linkage pattern of genetic markers flanking the GTM 1, we detected various GHD individuals in the meio-gynogenetic progeny of a hybrid (Figure 3a,b, Appendix A). These results substantiated that HR at GTM 1 resulted in the DNA fragment exchange between the Chinese GF and Japanese GF *ntl* promoter in a few eggs of the hybrid. 

### 2.4. Length Effect of GTM 1 on Methylation of Downstream ntl CpG Island 

To verify that the DNA fragment exchange between the Chinese GF and Japanese GF *ntl* alleles resulted in the methylation status conversion of the downstream Chinese GF and Japanese GF *ntl* CpG island during oogenesis, we examined the methylation status of the *ntl* CpG island in a GHD individual (named GHD 1) by BS. The genotype of GHD 1 is composed of a wild type Japanese GF *ntl* allele and a recombinant Chinese GF *ntl* allele. In the recombinant Chinese GF *ntl* allele, the upstream sequence of the Chinese GF *ntl* promoter, including the GTM 1, is replaced by the corresponding promoter sequence of the Japanese GF *ntl* allele (Figure 3a and Appendix A). In the eggs of GHD 1, all the examined Japanese GF-derived CpG island clones were unmethylated as observed in the eggs of the Japanese GF strain, the examined Chinese GF-derived CpG island clones were unmethylated or significantly hypomethylated (Figure 3c). These results substantiated that the wild type Japanese GF *ntl* alleles maintained the unmethylation status of the CpG island in the eggs of the hybrid, and the replacement of the Chinese GF upstream promoter sequence by the corresponding Japanese GF sequence resulted in the conversion of the downstream CpG island in the Chinese GF *ntl* allele from methylation to non-methylation status. These results also revealed that polymorphisms of a short GT microsatellite (GTM 3) and non-repetitive sequences in the region of the promoter downstream GTM 1 have no significant impact on the methylation of the CpG island, the *cis*-acting regulatory element involved in mediating the *ntl* CpG island methylation is at the upstream region of the *ntl* promoter. 

To determine whether the length of GTM 1 in the upstream region of the *ntl* promoter has an impact on the downstream CpG island methylation, we further examined the methylation status of the Chinese GF CpG island in another GHD individual (named GHD 2) and a gynogenetic homozygous individual by BS. GHD 2 contains a wild type Chinese GF *ntl* allele and a recombinant Chinese GF *ntl* allele, in which the long Chinese GF GTM 1 is replaced by the short Japanese GF GTM 1 (Figure 3b and Appendix A). In the eggs of GHD 2, about 50% of the sequenced clones was highly methylated and the other 50% was unmethylated or significantly hypomethylated (Figure 3d). In the eggs of the gynogenetic homozygous diploid individual, which contains the duplicated wild type Chinese GF *ntl* alleles, all the sequenced clones of the Chinese GF CpG island were highly methylated as observed in the eggs of the original Chinese GF strain. These results substantiated that wild type Chinese GF *ntl* alleles also maintained the methylation status of the CpG island in the eggs of hybrid progeny. The replacement of the long Chinese GF GTM 1 by the short Japanese GF GTM 1 resulted in the conversion of the CpG island in the Chinese GF allele from methylation to non-methylation status, indicating that GTM 1 in the upstream region of the *ntl* promoter can mediate methylation modification of the downstream CpG island in a length-dependent manner. Taken together, we can conclude that the heterozygous polymorphic GTM 1 can lead to the ASM of the downstream *ntl* CpG island in the eggs of the hybrid. 

### 2.5. ASM of the ntl CpG Island and ASE of ntl in Heterozygous Embryos 

We further examined whether the ASM of the *ntl* CpG island occurs during embryogenesis in the hybrid. The BS analysis showed that the *ntl* CpG island was highly methylated in the Chinese GF embryos and slightly methylated in the Japanese GF embryos at the four-day old stage (Figure 4a). In the heterozygous embryos generated by the reciprocal crossing of a Chinese GF homozygous male with a Japanese GF homozygous female or a Japanese GF homozygous male with Chinese GF homozygous female, the methylation level of the Chinese GF CpG island was generally higher than that of the Japanese GF CpG island as a whole at the four-day old stage (Figure 4a). Low proportions of the Chinese GF-derived CpG island clones and Japanese GF-derived CpG island clones also exchanged their methylation and non methylation states in both of the reciprocal hybrid embryos. The parental origin of the *ntl* alleles exhibited no significant effect on the methylation of the Chinese GF and Japanese GF CpG island. These results indicated that the Chinese and Japanese GF-derived *ntl* CpG island were allele-specifically methylated in the heterozygous embryos at the four-day old stage in a similar manner as observed in the eggs of the hybrid, suggesting that the GTM 1 might also mediate the ASM of the *ntl* CpG island during embryogenesis in a length-dependent manner. 

To determine whether the ASM of the CpG island can lead to the ASE of *the ntl* gene, we directly measured the expression of the Chinese and Japanese GF-derived *ntl* alleles in the reciprocal hybrid embryos with two SNP sites in the 3’-untranslated region (Figure 4b,c). By sequencing the RT-PCR products amplified from the heterozygous embryos and counting the Chinese and Japanese GF-derived transcript clones, we observed that the Japanese GF-derived transcripts was always much more abundant than Chinese GF-derived transcripts in the reciprocal hybrid embryos at the four-day old stage (Figure 4d,e). Of the sequenced clones in the reciprocal hybrid embryos, the ratio of the Chinese GF-derived transcript clones to the Japanese GF-derived transcript clones was four to 17 in one heterozygous embryos and one to 20 in the other heterozygous embryos. This indicated that the ASM of the CpG island was associated with the ASE of the *ntl* in the advanced embryos of the hybrid. Since previous studies have demonstrated that the methylation of the CpG island is involved in *ntl* repression in both goldfish and zebrafish [28,29], it is reasonable that the ASM of the CpG island resulted in the ASE of the *ntl* in the heterozygous embryos. 

## 3. Discussion

In this study, we generated various intraspecific goldfish heterozygotes and examined whether the length polymorphisms of GT microsatellites within the *ntl* promoter have impact on the non-imprinted ASM of the downstream CpG island during normal gametogenesis and embryogenesis. We found that the length polymorphisms of a long GT microsatellite within the *ntl* promoter could lead to the ASM of the downstream CpG island in both eggs and embryos of the hybrid, and polymorphisms of a short GT microsatellite and non-repetitive sequences in the upstream promoter region exhibited no significant effect on the methylation of the CpG island. We also observed that the ASM of the CpG island was associated with the ASE of the *ntl* in the heterozygous embryos at the late developmental stage. These results suggest that long polymorphic GT microsatellites within the gene promoter are a kind of *cis*-acting elements that can mediate the non-imprinted ASM of the local CpG island in the heterozygote of vertebrates.

During the process of genome activity, tandem-repeat microsatellite DNA sequences readily adopt non-Watson-Crick secondary structures including hairpins and slipped strands [34,35,36]. Moreover, the GT microsatellite sequences are known for their ability to form a left-handed helix structure [37]. The unusual secondary structures formed at long GT microsatellites may alter spacing of cooperative elements that function optimally at certain separation to mediate de novo methylase binding to the downstream CpG island. This may explain why length polymorphisms at a long GT microsatellite locus rather than at a short one can affect the methylation of the downstream CpG island. The length polymorphisms of non-repetitive sequences in the downstream promoter regions exhibited no effect on the *ntl* CpG island methylation, supporting the importance of the unusual secondary structure formed at a long GT microsatellite in regulating the methylation of the downstream CpG island. 

SNPs at CpG dinucleotides can create or delete CpG sites. The presence of heterozygous CpG SNPs in the surrounding local DNA sequences of the ASM region also can result in allele-specific differences in the density of CpGs and thereby influence the methylation propensity of neighboring non-polymorphic CpGs [6,7,10,11]. Genome-wide surveys of the non-imprinted ASM the human genome have observed that a significant fraction of the non-imprinted ASM regions outside of the CpG island are solely due to the presence of SNPs at dinucleotides [6]. In the present study, however, the insertion or deletion of two CpG sites in the Japanese GF CpG island has no detectable effect on methylation of the CpG island, suggesting that CpG SNPs within the CpG island is unlikely an important factor for the ASM of the CpG island at non-imprinting loci. 

Frequent HR occurs at the GTM 1 and the generated DNA fragment exchange at the upstream promoter region between the Chinese GF and Japanese GF *ntl* alleles during meiosis of the hybrid can result in the conversion of the CpG island in the Chinese GF allele from methylation to non-methylation status. Similar conversion of methylation status of the Chinese GF-derived CpG island also occurs in the embryonic cells of the hybrid, suggesting that frequent HR might occur at the GTM 1 region in a similar manner during mitosis. It is possible that the long GT microsatellite is prone to DNA double-strand breaks due to replication stalling and slippage during genome replication, and thereby capable to initiate HR in the process of division of both the germ and somatic cell. 

## 4. Materials and Methods

### 4.1. Animals and Samples Preparation

The Chinese goldfish (Chinese GF) strain (*Carassius auratus auratus*) and the Japanese goldfish (Japanese GF) strain (*Carassius auratus cuvieri*) used in our study are bisexual diploid goldfish strains. Both Chinese GF and Japanese GF were purchased from nearby farms and maintained in our laboratory in the breeding season according to the Guidelines for the Care and Use of Laboratory Animals of Zhejiang University (Zju 201306-1-11-060). The inter-strain specific hybrid was generated by crossing with a homozygous male individual from one strain and a homozygous female individual from the other strain. Sperms and eggs were obtained from breeding fish by gentle squeezing. Embryos were collected at the desired stage. Gynogenetic diploid progeny of the hybrid was obtained by activating the matured eggs of a female hybrid individual with genetic inactivated common carp sperm, and cold shocking the activated eggs at 0–2 °C for 15 min to inhibit the second polar body release as described previously [28]. All the samples were flash frozen in liquid nitrogen until genomic DNA or mRNA extraction. 

### 4.2. Bisulfite Sequencing (BS) of the ntl CpG Island 

Genomic DNA was converted by the bisulfite salt treatment according to the instruction of a CpGenomeTM DNA Modification Kit (Chemicon, Temecula, CA, USA). The primers used for amplifying the bisulfite converted sequence of the *ntl* CpG island were designed according to the sequence of the CpG island in the goldfish *ntl* promoter (GenBank accession number: KU870662). The sequences of the pair of primers were 5′-TTTTGTAATGGATTTTTGTGTAAGT-3′ (forward) and 5′-AACTCTTTAAAC TTACTCCATAACTC-3′ (reverse), PCR products were purified and cloned into pMD18-T Simple Vector (Takara, Tokyo, Iapan) and subjected to DNA sequencing. All the bisulfate converted sequences were compared with the original genomic sequence of the *ntl* CpG island and only those clones in which all the single cytosines (except at the CpG sites) were converted into thymines were used for statistical analysis. 

### 4.3. Methylation-Specific PCR (MSP) 

Genomic DNA of gametes digestion was performed using restriction enzymes Msp I and Hpa II. Both enzymes recognized the sequence CCGG, while not digested by the Hpa II when the CG dinucleotide cytosine was methylated. The reactions were carried out for 16 h at 37 °C in a volume of 20 µL and followed by PCR amplification with specific primers: 5′-ATTCTTTCCGCTTGTGGCGACAGATTAT-3′ (forward); 5′-GTGCACTAGATCCTGTGTGGCCGCG-3′ (reverse), which were designed according to the sequence of the CpG island in the goldfish *ntl* promoter region (GenBank accession number: KU870662). 

### 4.4. Measurements of Expression Differences between Chinese GF and Japanese GF ntl Alleles in the Heterozygous Embryos 

Two single nucleotide polymorphism sites (SNPs) were identified at the 3′-untranslated region (3′-UTR) of the goldfish *ntl* mRNA and used in distinguishing the Chinese and Japanese GF-derived alleles transcripts. The expression differences between the two parental *ntl* alleles in the heterozygous embryos were measured by counting and comparing the number of sequenced clones derived from the Chinese GF or Japanese GF derived alleles. The primers used in amplifying the (3′-UTR) of goldfish *ntl* mRNA fragment containing the two SNPs were 5’-GCCATACATTTTGGGGCAATTATAGTGT-3′ (forward) and 5′-TTCAAAGA GCCCAACAAATACAAATAGAT-3′ (reverse), designed according to the full length goldfish *ntl* mRNA sequence in the gene bank (GenBank accession number: EU549782). 

## Figures and Tables

**Figure 1 ijms-20-03923-f001:**
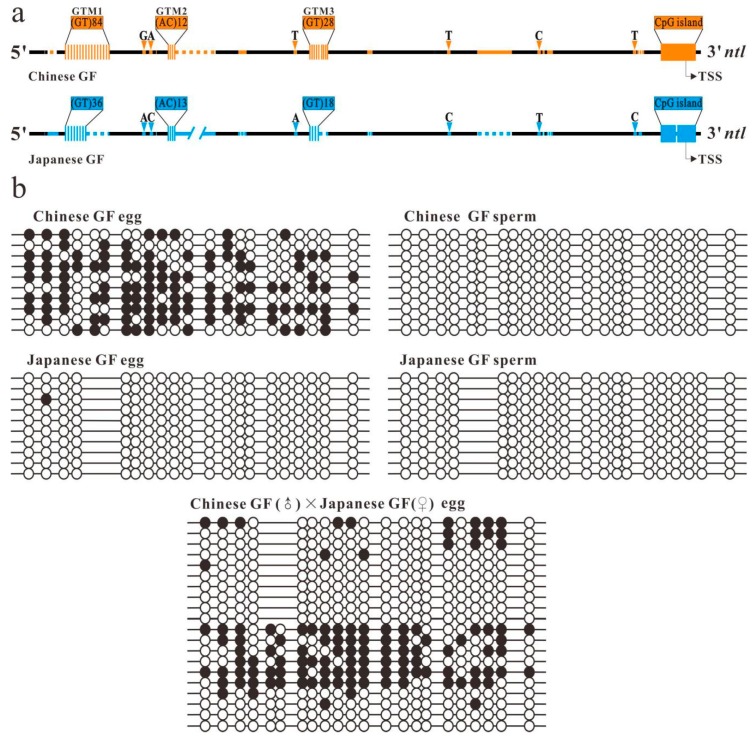
Sequence divergence of the *no tail (ntl)* promoter is associated with methylation variation of the linked CpG island. (**a**) Comparison of Chinese goldfish (GF) (*C. auratus auratus*) and Japanese GF (*C. auratus cuvieri*) *ntl* promoter in the intraspecific hybrid. Black lines represent the conserved sequence regions. Orange and blue clusters of vertical bars, lines, dotted lines, arrowheads and boxes represent the paternal and maternal specific GT microsatellites (GTMs), inserted sequences, deleted sequences, single-nucleotide polymorphism (SNP) sites and CpG island, respectively. TSS indicates the transcription start site. The number represents the sequence length from the GTM1 to the CpG island. (**b**) Bisulfite sequencing results of the *ntl* promoter CpG island in the gametes of a heterozygous individual and the parental controls. The methylated and unmethylated CpG sites are represented by filled circles and open circles, respectively. Each line represents an examined clone. The Chinese and Japanese GF-derived *ntl* CpG island clones in the hybrid can be distinguished by the insertion/deletion genetic marker.

**Figure 2 ijms-20-03923-f002:**
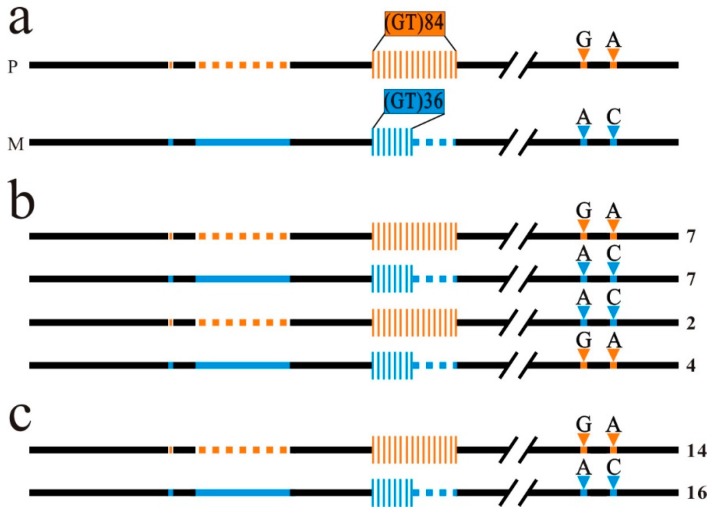
Homologous recombination occurs at GTM 1 region in the eggs of a heterozygous individual. (**a**) The paternal and maternal genetic makers flanking the GTM 1. P and M indicate the paternal and maternal alleles. (**b**) The linkage patterns of the paternal and maternal genetic makers flanking the GTM 1 in the eggs of a hybrid. (**c**) No homologous recombination clone was detected in the PCR products amplified from the equivalent mixture of homozygous paternal and maternal genomic DNA. Black lines represent the conserved regions. Orange and blue symbols represent the paternal and maternal genetic markers as indicated in Figure 2. Number of sequenced clones is indicated at the right. Intra-allelic recombination clones with contraction or expansion of GTM 1 detected in all the experimental and control samples were eliminated in the statistical data.

**Figure 3 ijms-20-03923-f003:**
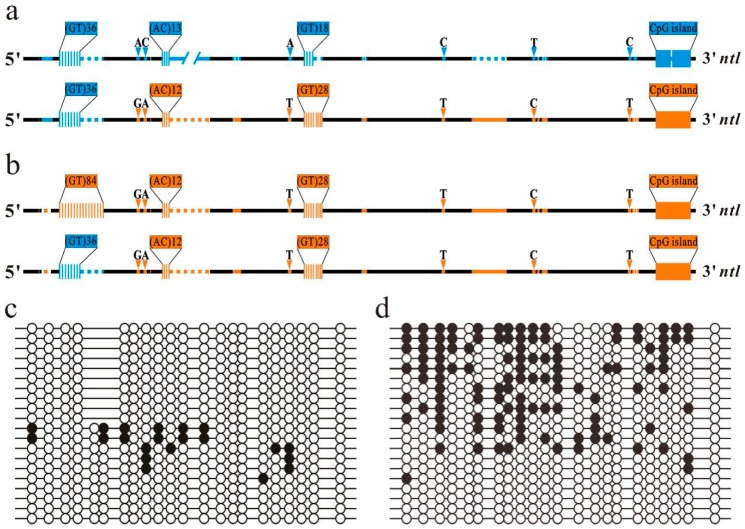
The length of GTM 1 in the upstream region of the *ntl* promoter has an impact on the methylation of the downstream CpG island. (**a**) The wild type Japanese GF *ntl* allele promoter and the recombinant Chinese GF *ntl* allele promoter in the GHD 1. (**b**) The wild type Chinese GF *ntl* allele promoter and the recombinant Chinese GF *ntl* allele promoter in the GHD 2. The black represents the conserved sequence regions. Orange and blue symbols represent the Chinese GF and Japanese GF genetic markers as indicated in Figure 2. (**c**) and (**d**) Bisulfite sequencing results of the *ntl* promoter CpG island in the eggs of the GHD 1 and 2, respectively. The methylated and unmethylated CpG sites are represented by filled circles and open circles, respectively. Each line represents an examined clone. The Chinese and Japanese GF-derived *ntl* CpG island clones in GHD 1 can be distinguished with the insertion/deletion genetic marker.

**Figure 4 ijms-20-03923-f004:**
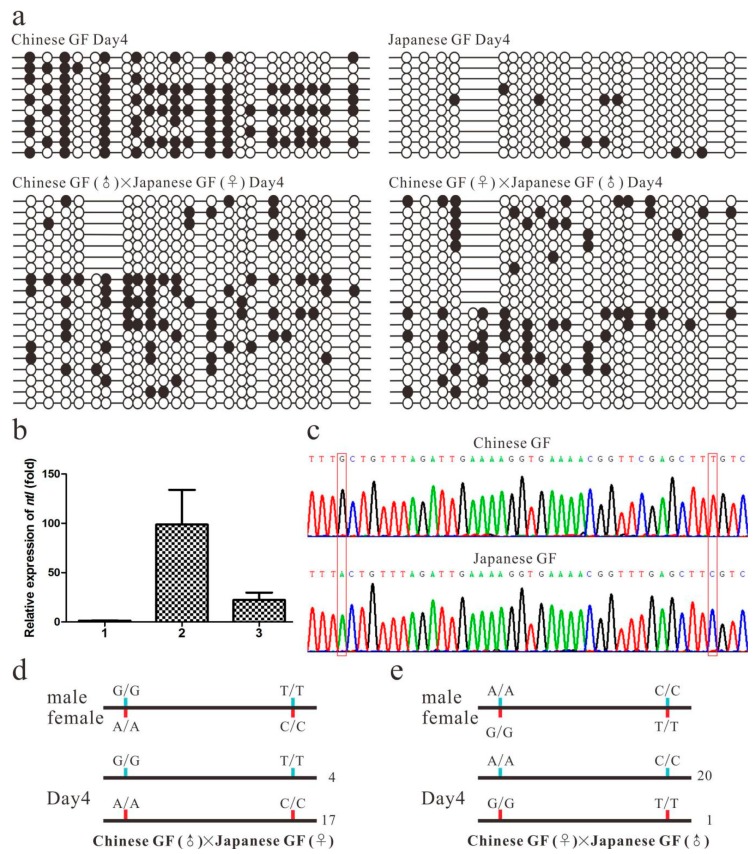
ASM and ASE of the *ntl* allele in the advanced embryos of the hybrid. (**a**) The methylation status of the Chinese GF (*C. auratus auratus*) and Japanese GF (*C. auratus cuvieri*) *ntl* CpG island in control strains and reciprocal crosses at the late embryonic stage. (**b**) The expression level of *ntl* gene in control strains and reciprocal crosses at the late embryonic stage. 1 represents Chinese GF, 2 represents Japanese GF, 3 represents the mix of both reciprocal crosses. (**c**) Two SNP sites in the 3’-untranslated region of the Chinese GF and Japanese GF *ntl* mRNA. The red boxes indicate the SNP sites. (**d**) and (**e**) Transcription difference of Chinese and Japanese GF-derived *ntl* alleles in the advanced embryos of both reciprocal crosses. The red and blue color lines indicate female- and male-derived clones, respectively. More than 20 clones of RT-PCR products were sequenced. Numbers at the right of each horizontal line in C and D indicate the number of clones identified in the sequenced clones. The Chinese and Japanese GF-derived *ntl* CpG island clones in the hybrid can be distinguished with the insertion/deletion genetic marker.

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
