# Peer review of "A Long Polymorphic GT Microsatellite within a Gene Promoter Mediates Non-Imprinted Allele-Specific DNA Methylation of a CpG Island in a Goldfish Inter-Strain Hybrid"

_ijms, 2019, doi:10.3390/ijms20163923_

Round 1

Reviewer 1 Report

This is a very nice paper. The science is well done, the results are interesting, experiment and results are nicely documented and all data are very convincing. I recommend publication of this work after some minor corrections.

Many places: “CGF ntl promoter”, “ntl promoter” and “ntl CpG island” should both be preceded by “The”. Check and correct global.

Line 33: Include ref. 4 as well.

Line 43 and many other places: It must read “A recent study…”, “Previous study”, etc. The manuscript should be searched for the phrase “study” and corrected where needed.

Line 44: “These findings stimulated us…”

Line 48: “…a key developmental regulatory gene” move here “in zebrafish and goldfish” from the end of the sentence.

Line 68: It should read “The” Chinese goldfish and “The” Japanese goldfish… not “A”

Line 115 and 116: Remove “much” in both lines.

Fig. 1: Indicate in a the region that is sequenced in b. Are these the arrows marked by bp sizes? Please clarify.

Line 145: “To prove that the …”

Line 152: Correct: “After sexual maturity way reached, we”. Why is the beginning of the sentence in italics?

Line 163: “to speculate about the following”

Line 180: “during the PCR reaction”

Line 265: “much more abundant than…”

Author Response

Rviewer 1#:

This is a very nice paper. The science is well done, the results are interesting, experiment and results are nicely documented and all data are very convincing. I recommend publication of this work after some minor corrections.

Many places: “CGF ntl promoter”, “ntl promoter” and “ntl CpG island” should both be preceded by “The”. Check and correct global.

ResponceThanks for your reminding. Revised as asked.

Line 33: Include ref. 4 as well.

ResponceThanks for your reminding. Revised as asked.

Line 43 and many other places: It must read “A recent study…”, “Previous study”, etc. The manuscript should be searched for the phrase “study” and corrected where needed.

Revised as asked.

Line 44: “These findings stimulated us…”

ResponceRevised as asked.

Line 48: “…a key developmental regulatory gene” move here “in zebrafish and goldfish” from the end of the sentence.

ResponceRevised as asked.

Line 68: It should read “The” Chinese goldfish and “The” Japanese goldfish… not “A”

ResponceRevised as asked.

Line 115 and 116: Remove “much” in both lines.

ResponceRevised as asked.

Fig. 1: Indicate in a the region that is sequenced in b. Are these the arrows marked by bp sizes? Please clarify.

ResponceRevised as asked.

Line 145: “To prove that the …”

ResponceRevised as asked.

Line 152: Correct: “After sexual maturity way reached, we”. Why is the beginning of the sentence in italics?

ResponceRevised as asked.

Line 163: “to speculate about the following”

ResponceRevised as asked.

Line 180: “during the PCR reaction”

ResponceRevised as asked.

Line 265: “much more abundant than…”

ResponceRevised as asked.

Reviewer 2 Report

My major concern with this paper is the title and abstract that imply a different result than that obtained.

I recommend a title such as:  A long polymorphic GT microsatellite within a gene promoter mediates strain-specific DNA methylation of a CpG island in a goldfish interspecies hybrid.

The abstract should be modified accordingly. Currently the abstract refers to “gene promoters” and “CpG islands” but only one promoter and one CpG island of one gene was studied.  The final sentence of the abstract is too generalised and too strong.

Given that these fish are inter-strain hybrids, I'm not so sure that "allele specific" is the correct terminology. Surely the differences are strain specific (can you check with an appropriate expert).  The consequences are equally interesting, but for clarity it might be preferable to refer to strain-specific.

Define the ntl gene using standard nomenclature (e.g., NCBI). The current NCBI gene name is Gene symbol is LOC113119157 and the gene description is brachyury protein homolog A.  “No tail” is a name of the protein although the preferred name is brachyury protein homolog A

Instead of using JGF and CGF initialisms in the text, it would be much easier if they referred to strains as Chinese and Japanese (or alternatively auratus and cuvieri). At present the text tends to refer JGF and CGF, whereas the figures use the Latin names.  A single easy naming system would be much better.  The benefit of readability would outweigh the slightly longer text.

Assuming a modified title, the results appear to support the main conclusions. The experiments are thorough and convincing.

The text refers to GTM 1, 2 and 3, referring the reader to Figure 1a in which the GT microsatellites are not labelled.

It would help if the authors explicitly stated that the studied fish were F1 hybrids.

Homologous recombination was stated to occur in a “small proportion” of eggs.  This needs to be expressed numerically (was it 1 of 3, or 3 of 100, or 10 of a 1000?).  

Line 252 “always significant” needs more explanation. What significance test was used? When I compared the proportions of methylation in the Chinese vs Japanese alleles using a T-test, the difference was not significant.

Spelling errors (please use spell checker)

18  hyterozygotes

45. eucaryotes

66 theses hyterozygotes

79  2.2. Bisulfite sequencing (BS) of ntl CpG islands  (only one CpG island)

86 bisulfate

283 4. DISCCUSION

306 dineucleotides

Supplement spelling errors

Firgure

Bilsufate

Heterozugous

Aliment

Sperms (sperm is always singular).

Author Response

Reviewer 2#

My major concern with this paper is the title and abstract that imply a different result than that obtained.

I recommend a title such as:  A long polymorphic GT microsatellite within a gene promoter mediates strain-specific DNA methylation of a CpG island in a goldfish interspecies hybrid. The abstract should be modified accordingly. Currently the abstract refers to “gene promoters” and “CpG islands” but only one promoter and one CpG island of one gene was studied.  The final sentence of the abstract is too generalised and too strong. Given that these fish are inter-strain hybrids, I'm not so sure that "allele specific" is the correct terminology. Surely the differences are strain specific (can you check with an appropriate expert). The consequences are equally interesting, but for clarity it might be preferable to refer to strain-specific.

Responce: Thanks for your suggestion. The difference between the two strains of goldfish was only in alleles, but the reproduction of the hybrid progenies was normal. Therefore, appearance the phrase of “allele-specific DNA methylation” is not inappropriate in scientific definition. To make the expression more accurate, it is defined using the phrase of in a goldfish inter-strain hybridin the revised draft on the basis of your advice. In addition, some errors occurred in the abstract have beed revised and the final sentence of the abstract also have been changed.

Define the ntl gene using standard nomenclature (e.g., NCBI). The current NCBI gene name is Gene symbol is LOC113119157 and the gene description is brachyury protein homolog A.  “No tail” is a name of the protein although the preferred name is brachyury protein homolog A

Responce: Thanks for your reminding. Although the gene description is brachyury protein homolog A, at the same time it is also pointed out that also known as ntl. So we think they don't make much difference.

Instead of using JGF and CGF initialisms in the text, it would be much easier if they referred to strains as Chinese and Japanese (or alternatively auratus and cuvieri). At present the text tends to refer JGF and CGF, whereas the figures use the Latin names.  A single easy naming system would be much better.  The benefit of readability would outweigh the slightly longer text.Assuming a modified title, the results appear to support the main conclusions. The experiments are thorough and convincing.

Responce: Thanks for your reminding. We unify the naming of the two strains in this paper, also including figures and supplementary data.

The text refers to GTM 1, 2 and 3, referring the reader to Figure 1a in which the GT microsatellites are not labelled.

Responce: Thanks for your reminding. Revised as asked.

It would help if the authors explicitly stated that the studied fish were F1 hybrids.

Responce: In this study, we screened two bisexual goldfish strains with different genetic backgrounds. More importantly, inter-strain specific hybrid of F1 was generated to explain our concern on the role of GTM in methylation modification.

Homologous recombination was stated to occur in a “small proportion” of eggs.  This needs to be expressed numerically (was it 1 of 3, or 3 of 100, or 10 of a 1000?).  

Responce: Thanks for your reminding. We have added the number that represents the proportion of HR in hybrid egg.

Line 252 “always significant” needs more explanation. What significance test was used? When I compared the proportions of methylation in the Chinese vs Japanese alleles using a T-test, the difference was not significant.

Responce: Thanks for your suggestion. It may be an exaggeration to use “always significantly” here, but we can confirm that the level of methylation of the Chinese GF--derived clones are generally higher than that of the Japanese GF--derived clones in hybrid embryos, and according to your suggestion, we have made appropriate changes to the statement.

Spelling errors (please use spell checker)

18  hyterozygotes

eucaryotes

66 theses hyterozygotes

79  2.2. Bisulfite sequencing (BS) of ntl CpG islands  (only one CpG island)

86 bisulfate

283 4. DISCCUSION

306 dineucleotides

Supplement spelling errors

Firgure

Bilsufate

Heterozugous

Aliment

Sperms (sperm is always singular).

Response: All spelling errors have been revised.